CERN-TH-2022-095

# Holographic thermal correlators from supersymmetric instantons

Matthew Dodelson$^a$, Alba Grassi$^{a,b}$, Cristoforo Iossa$^{c,d}$, Daniel Panea Lichtig$^{c,d}$, and Alexander Zhiboedov$^a$

$^a$*CERN, Theoretical Physics Department, CH-1211 Geneva 23, Switzerland*
$^b$*Section de Mathématiques, Université de Genève, 1211 Genève 4, Switzerland*
$^c$*SISSA, via Bonomea 265, 34136 Trieste, Italy and*
$^d$*INFN, sezione di Trieste, via Valerio 2, 34127 Trieste, Italy*

We present an exact formula for the thermal scalar two-point function in four-dimensional holographic conformal field theories. The problem of finding it reduces to the analysis of the wave equation on the AdS-Schwarzschild background. The two-point function is computed from the connection coefficients of the Heun equation, which can be expressed in terms of the Nekrasov-Shatashvili partition function of an $SU(2)$ supersymmetric gauge theory with four fundamental hypermultiplets. The result is amenable to numerical evaluation upon truncating the number of instantons in the convergent expansion of the partition function. We also examine it analytically in various limits. At large spin the instanton expansion of the thermal two-point function directly maps to the light-cone bootstrap analysis of the heavy-light four-point function. Using this connection, we compute the OPE data of heavy-light double-twist operators. We compare our prediction to the perturbative results available in the literature and find perfect agreement.

## CONTENTS

## I. INTRODUCTION

In this paper we study the thermal two-point function in a holographic four-dimensional CFT[1] [2–4] using techniques coming from four-dimensional supersymmetric gauge theories [5–9]. Above the Hawking-Page transition [10] this observable is computed by studying the wave equation on the AdS-Schwarzschild background [11]. Thermal correlation functions contain a wealth of fascinating physics related to the richness of the black hole geometry. For example, two-point functions encode the transport properties of the system, see e.g. [12, 13], the approach to equilibrium [14], as well as chaotic dynamics via pole-skipping [15, 16]. Thermal four-point functions serve as an important diagnostic of quantum chaos [17, 18]. Thermal correlators have also been used to formulate a version of the information paradox [19], as well as to look for subtle signatures of the black hole singularity [20–23].

Finite temperature dynamics of CFTs is particularly rich in $d > 2$, where propagation of energy is not fixed by symmetries. On the gravity side, this is related to the presence of a propagating graviton in the spectrum of the theory, namely gravity waves.[2] On the field theory side, it is due to the fact that conformal symmetry is finite-dimensional in $d > 2$. This richness comes at a price

———

[1] We consider a finite-temperature CFT on the sphere, $S_\beta^1 \times S^3$, and on the plane, $S_\beta^1 \times \mathbb{R}^3$. The former is related to the black hole geometry, and the latter to the black brane. The requirement of being holographic implies a large CFT central charge ($c_T \gg 1$), and a large gap in the spectrum of higher spin single trace operators ($\Delta_{\text{gap}} \gg 1$) [1].

[2] Another characteristic feature of black holes in $d > 2$ is the existence of stable orbits [24–26].

that even for the simplest finite temperature observables no explicit solutions are available in $d > 2$.[3]

In this paper we provide the first example of such an explicit result. The thermal two-point function is computed by studying the wave equation on the black hole background [29–31]. This equation is of the Heun type [32–34], and the retarded two-point function is given in terms of its connection coefficients. Starting with [9], a growing body of problems of this class have been solved using the connection to Seiberg-Witten theory and more precisely the Nekrasov-Shatashvili (NS) functions. These ideas have been applied to the study of black hole perturbation theory in [35–42][4]. In particular this connection allows us to express the thermal two-point function in terms of the NS free energy [9] of an $SU(2)$ gauge theory with four fundamental hypermultiplets, and to study some of its basic properties both analytically and numerically. One particularly interesting regime is the large spin limit, where the exact formula produces the solution to the heavy-light light-cone bootstrap [46, 47]. We reproduce the available perturbative results from the literature [26, 48–61] and make new predictions.

## II. HOLOGRAPHIC TWO-POINT FUNCTION AT FINITE TEMPERATURE

### A. Black hole

We consider a holographic conformal field theory at finite temperature. Above the Hawking-Page transition [10], this theory is dual to a black hole in AdS [11]. In this paper we will specialize to the case of $AdS_5$, where the black hole metric is

$$ds^2 = -f(r)\,dt^2 + f(r)^{-1}\,dr^2 + r^2 d\Omega_3^2. \tag{1}$$

Setting the AdS radius to 1, the redshift factor takes the form

$$f(r) = r^2 + 1 - \frac{\mu}{r^2}$$
$$\equiv \left(1 - \frac{R_+^2}{r^2}\right)(r^2 + R_+^2 + 1), \tag{2}$$

where the Schwarzschild radius is given by

$$R_+ = \sqrt{\frac{\sqrt{1 + 4\mu} - 1}{2}}. \tag{3}$$

The dimensionless parameter $\mu$ is related to the black hole mass $M$ by

$$\mu = \frac{8G_N M}{3\pi}\ . \tag{4}$$

---

[3]Here we refer to the black hole phase. For the thermal AdS phase some explicit results exist [27]. They are also available in $d \le 2$, see e.g. [28].
[4]See also [43–45] for a different approach based on Painlevé equations.

We are interested in the two-point function of a scalar operator $\mathcal{O}(x)$ with dimension $\Delta$, dual to a massive scalar $\phi$ in the bulk with mass [62]

$$m = \sqrt{\Delta(\Delta - 4)}. \tag{5}$$

In order to compute this two-point function, we need to solve the wave equation on the black hole background,

$$(\Box - m^2)\phi = 0. \tag{6}$$

Expanding the solution into Fourier modes, we have

$$\phi(t, r, \Omega) = \int d\omega \sum_{\ell, \vec{m}} e^{-i\omega t} Y_{\ell\vec{m}}(\Omega)\psi_{\omega\ell}(r). \tag{7}$$

Our conventions for spherical harmonics $Y_{\ell\vec{m}}$ can be found in Appendix A of [21]. The wave equation then takes the form (see [63] and references there)

$$\left(\frac{1}{r^3}\partial_r(r^3 f(r)\partial_r) + \frac{\omega^2}{f(r)} - \frac{\ell(\ell + 2)}{r^2} - \Delta(\Delta - 4)\right)\psi_{\omega\ell} = 0. \tag{8}$$

We are interested in the retarded Green's function, and therefore we impose ingoing boundary conditions on the solution $\phi$ at the horizon,

$$\psi_{\omega\ell}^{\text{in}}(r) = (r - R_+)^{-\frac{i\omega}{2}\frac{R_+}{2R_+^2+1}} + \dots \tag{9}$$

The solution $\psi^{\text{in}}$ behaves near the AdS boundary $r \to \infty$ as

$$\psi_{\omega\ell}^{\text{in}}(r) = \mathcal{A}(\omega, \ell)(r^{\Delta-4} + \dots) + \mathcal{B}(\omega, \ell)(r^{-\Delta} + \dots). \tag{10}$$

The two-point function is then the ratio of the response $\mathcal{B}(\omega, \ell)$ to the source $\mathcal{A}(\omega, \ell)$ [29],

$$G_R(\omega, \ell) = \frac{\mathcal{B}(\omega, \ell)}{\mathcal{A}(\omega, \ell)}. \tag{11}$$

Our conventions for the thermal two-point function in the CFT dual are collected in Appendix A.

The wave equation takes a particularly convenient form under the transformations

$$z = \frac{r^2}{r^2 + R_+^2 + 1}\ , \tag{12}$$

$$\psi_{\omega\ell}(r) = \left(r^3 f(r)\frac{dz}{dr}\right)^{-1/2}\chi_{\omega\ell}(z)\ . \tag{13}$$

We then obtain Heun's differential equation in normal form,

$$\left(\partial_z^2 + \frac{\frac{1}{4} - a_1^2}{(z-1)^2} - \frac{\frac{1}{2} - a_0^2 - a_1^2 - a_t^2 + a_\infty^2 + u}{z(z-1)}\right.$$
$$\left. + \frac{\frac{1}{4} - a_t^2}{(z-t)^2} + \frac{u}{z(z-t)} + \frac{\frac{1}{4} - a_0^2}{z^2}\right)\chi_{\omega\ell}(z) = 0. \tag{14}$$

Here the horizon is at $z = t$ and the AdS boundary is at $z = 1$.

In (14) we introduced a set of parameters that acquire a natural interpretation in the context of gauge theory that we discuss in the next section. They are defined in Table I.

| Gauge theory | $t$ | $a_0$ | $a_t$ | $a_1$ | $a_\infty$ |
|---|---|---|---|---|---|
| Black hole | $\frac{R_+^2}{2R_+^2+1}$ | $0$ | $\frac{i\omega}{2}\frac{R_+}{2R_+^2+1}$ | $\frac{\Delta-2}{2}$ | $\frac{\omega}{2}\frac{\sqrt{R_+^2+1}}{2R_+^2+1}$ |

TABLE I. Map from gauge theory to the black hole wave equation parameters.

Finally, $u$ is given by

$$u = -\frac{\ell(\ell+2) + 2(2R_+^2+1) + R_+^2\Delta(\Delta-4)}{4(R_+^2+1)}$$
$$+ \frac{R_+^2}{1+R_+^2}\frac{\omega^2}{4(2R_+^2+1)}. \qquad (15)$$

The purely ingoing solution behaves near the black hole horizon as

$$\chi^{\text{in}}_{\omega\ell}(z) = (t-z)^{\frac{1}{2}-a_t} + \dots . \qquad (16)$$

Close to the AdS boundary it takes the form

$$\chi^{\text{in}}_{\omega\ell}(z) \propto \mathcal{A}(\omega,\ell)\Big(\frac{1-z}{1+R_+^2}\Big)^{\frac{1}{2}-a_1} \qquad (17)$$
$$+ \mathcal{B}(\omega,\ell)\Big(\frac{1-z}{1+R_+^2}\Big)^{\frac{1}{2}+a_1} + \dots .$$

The solutions to Heun's equation are known as Heun functions, see e.g. [33], and these can be written as an infinite series expanded around one of the singular points $z = 0, t, 1, \infty$. The problem of finding *the response function* (11) therefore reduces to finding the so-called *connection formulae* for the Heun function which express a given solution around one singular point (16) in terms of the basis of solutions around another singular point (17). The corresponding connection coefficients were computed explicitly in [36], [5] and we use these results in the present paper.

### B. Black brane

The black brane is dual to CFT on $S^1 \times \mathbb{R}^3$, and can be obtained by taking the high-temperature limit $T \to \infty$ of the black hole, while keeping $\frac{\omega}{T} \equiv \hat{\omega}$ and $\frac{\ell}{T} \equiv |\mathbf{k}|$ fixed.

---

[5]See also [64–69] for explicit relations between NS functions and the Heun equation.

Here $\hat{\omega}$ and $\mathbf{k}$ are the dimensionless energy and three-momentum of the resulting theory on $S^1 \times \mathbb{R}^3$ in units of temperature. Recall that for the AdS-Schwarzschild black hole [11]

$$T = \frac{1}{\sqrt{2}\pi}\sqrt{\frac{1+4\mu}{\sqrt{1+4\mu}-1}}, \qquad (18)$$

and the high-temperature limit corresponds to $\mu \to \infty$.

In this way we get the map between the gauge theory and gravity parameters for the black brane (to avoid clutter we switch from $\hat{\omega}$ to $\omega$), see Table II.

| Gauge theory | $t$ | $a_0$ | $a_t$ | $a_1$ | $a_\infty$ |
|---|---|---|---|---|---|
| Black brane | $\frac{1}{2}$ | $0$ | $\frac{i\omega}{4\pi}$ | $\frac{\Delta-2}{2}$ | $\frac{\omega}{4\pi}$ |

TABLE II. Map from gauge theory to the black brane wave equation parameters.

For $u$ the relation takes the following form,

$$u = \frac{\omega^2 - 2\mathbf{k}^2}{8\pi^2} - \frac{1}{4}(\Delta-2)^2. \qquad (19)$$

Finally, we define the two-point function as follows,

$$G_R^{\text{brane}}(\omega, |\mathbf{k}|) = \lim_{T\to\infty}\frac{G_R(\omega T, |\mathbf{k}|T)}{T^{4a_1}}, \qquad (20)$$

see Appendix B for the detailed derivation.

### III. EXACT THERMAL TWO-POINT FUNCTION

Heun's equation coincides with the quantum Seiberg-Witten curve describing the gauge theory with four flavors ($N_f = 4$) and therefore it can be solved exactly using the Nekrasov-Shatashvili (NS) functions [9]. Another way of understanding this connection is by using the AGT correspondence and the fact that Heun's equation corresponds to the semiclassical limit of the BPZ equation satisfied by the five-point function with one degenerate insertion, see for instance [65–68]. For a review and a detailed list of references see [71].

Let us review the basic idea behind the exact solution of the connection problem. We consider a five-point function in the Liouville theory where one of the fields has been analytically continued to have degenerate quantum numbers. This five-point function satisfies the BPZ equation, which expresses the shortening of the Verma module of the degenerate field [72]. The BPZ equation reduces to the Heun equation in the semi-classical (large central charge) limit of the Liouville theory. The four singular points in the Heun equation correspond to insertions of the four operators (the fifth operator being the degenerate field). Crossing symmetry of the five-point function

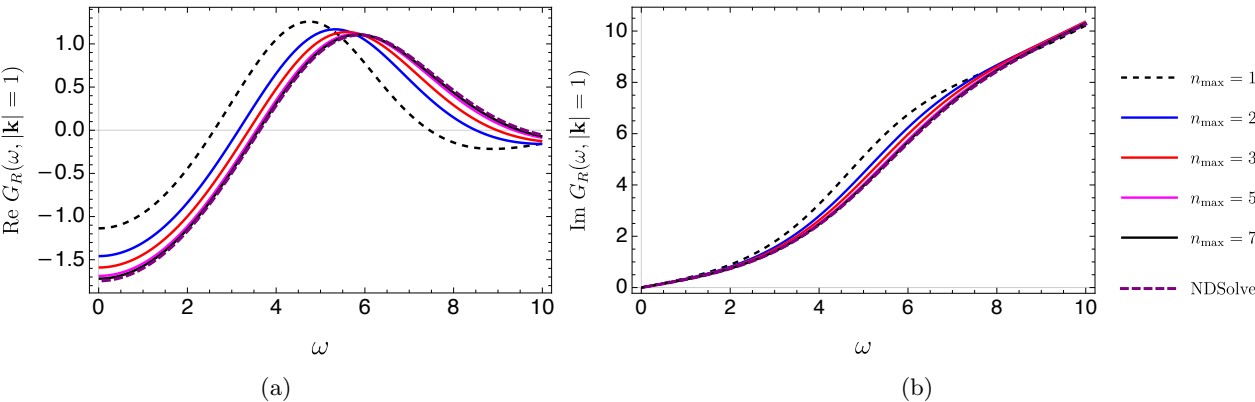

FIG. 1. We plot the retarded two-point function $G_R^{\text{brane}}(\omega, |\mathbf{k}|)$, given by (26) and (28), for $|\mathbf{k}| = 1$, $\Delta = 5/2$, as a function of $\omega$ and the maximal number of instantons $n_{\max}$ in the truncated sum (29). a) The real part of the retarded two-point function $\text{Re } G_R^{\text{brane}}(\omega, 1)$. b) The imaginary part of the retarded two-point function $\text{Im } G_R^{\text{brane}}(\omega, 1)$. We set $T = 1$. We also compare our results with the direct numerical solution of the differential equation (we used NDSolve in Mathematica), see e.g. [70], and find beautiful agreement between the two methods. An analogous plot can be generated for the $|\mathbf{k}|$-dependence as well, and again we observed perfect agreement between our formulas and the direct numerical solution of the differential equation.

leads to crossing relations between the Virasoro blocks in different OPE channels. In the semi-classical limit these descend to the connection formulae for the solutions of the Heun equation. Thanks to the DOZZ formula [73, 74] the three-point functions that enter the crossing relations are explicitly known. Similarly, via the AGT correspondence [8] the relevant Virasoro blocks are expressed in terms of the partition function of the four-dimensional gauge theory which enters our final result. On the gauge theory side, the semi-classical limit corresponds to the so-called Nekrasov-Shatashvili limit [9]. The resulting expression for the connection coefficients can be found in [36].

From the gauge theory point of view, the parameters $a_0, a_1, a_t, a_\infty$ are related to the masses of the hypermultiplets, $t \sim e^{-1/g_{\text{YM}}^2}$ is the instanton counting parameter, and $u$ parameterizes the moduli space of vacua. The latter is related to the VEV $a$ of the scalar in the vector multiplet via the (quantum) Matone relation [75, 76]

$$u = -a^2 + a_t^2 - \frac{1}{4} + a_0^2 + t\partial_t F, \tag{21}$$

where $F$ is the instanton part of the NS free energy defined in (C2). The dictionary (21) requires a careful treatment close to the points $2a = \mathbb{Z}$, where the NS function exhibits non-analyticity, see e.g. [77, 78]. We leave a more detailed discussion of this region for future work.

In particular this hidden connection between Heun's equation and supersymmetric gauge theory makes it possible to compute the connection coefficients $\mathcal{A}$ and $\mathcal{B}$ in (10) using the NS free energy, as done in [36].

Let

$$\chi_{\omega\ell}^{(t),\text{in}}(z) = (t - z)^{\frac{1}{2} - a_t} + \dots \tag{22}$$

be the ingoing solution of the wave equation (14) at the

horizon ($z \sim t$) and let

$$\chi_{\omega\ell}^{(1),\pm}(z) = (1 - z)^{\frac{1}{2} \pm a_1} + \dots \tag{23}$$

be the two independent solutions at infinity ($z \sim 1$). The connection formula reads

$$\chi_{\omega\ell}^{(t),\text{in}}(z) = \sum_{\theta'=\pm} \left( \sum_{\sigma=\pm} \mathcal{M}_{-\sigma}(a_t, a; a_0)\mathcal{M}_{(-\sigma)\theta'}(a, a_1; a_\infty) \right.$$
$$\left. t^{\sigma a} e^{-\frac{\sigma}{2}\partial_a F} \right) t^{\frac{1}{2} - a_0 - a_t}(1 - t)^{a_t - a_1}$$
$$e^{\frac{1}{2}\left(-\partial_{a_t} - \theta'\partial_{a_1}\right)F}\chi_{\omega\ell}^{(1),\theta'}(z) , \tag{24}$$

where

$$\mathcal{M}_{\theta\theta'}(\alpha_0, \alpha_1; \alpha_2) = \tag{25}$$
$$\frac{\Gamma(-2\theta'\alpha_1)}{\Gamma\left(\frac{1}{2} + \theta\alpha_0 - \theta'\alpha_1 + \alpha_2\right)} \frac{\Gamma(1 + 2\theta\alpha_0)}{\Gamma\left(\frac{1}{2} + \theta\alpha_0 - \theta'\alpha_1 - \alpha_2\right)},$$

and $F$ is the instanton part of the NS free energy defined in (C2).

The exact formula for the retarded two-point function (11) then reads

$$G_R(\omega, \ell) = \left(1 + R_+^2\right)^{2a_1} e^{-\partial_{a_1} F}$$
$$\frac{\sum_{\sigma'=\pm} \mathcal{M}_{-\sigma'}(a_t, a; a_0)\mathcal{M}_{(-\sigma')+}(a, a_1; a_\infty)t^{\sigma'a}e^{-\frac{\sigma'}{2}\partial_a F}}{\sum_{\sigma=\pm} \mathcal{M}_{-\sigma}(a_t, a; a_0)\mathcal{M}_{(-\sigma)-}(a, a_1; a_\infty)t^{\sigma a}e^{-\frac{\sigma}{2}\partial_a F}}$$

$$\tag{26}$$

where the parameters $t, a_0, a_t, a_1, a_\infty, u$ were defined in terms of $\omega, \ell$ and the mass of the black hole $\mu$ in Table I and equation (15). The instanton part of the free energy $F$ depends on all parameters, $F(t, a, a_0, a_t, a_1, a_\infty)$. Finally, we can eliminate $a$ from the problem using the Matone relation (21). In this way the right hand side of (26) is fully fixed in terms of $\omega, \ell$ and $\mu$.

Based on general grounds, $G_R(\omega, \ell)$ should be analytic in the upper half-plane (causality), it satisfies Im $G_R(\omega, \ell) = -$Im $G_R(-\omega, \ell)$ (KMS), and finally Im $G_R(\omega, \ell) \geq 0$ for $\omega > 0$ (unitarity), see e.g. appendix B in [79]. In fact from the standard dispersive representation of $G_R(\omega, \ell)$ it follows that

$$[G_R(-\omega, \ell)]^* = G_R(\omega, \ell), \quad \omega \in \mathbb{R}. \tag{27}$$

In this paper we mostly limit our analysis to $\omega \in \mathbb{R}$ and it is easy to check that (26) indeed satisfies (27). The argument for this goes as follows. First, we notice that for real $\omega$ and $\ell$, the relevant $a$ is either purely imaginary or purely real. Second, we notice that (26) is invariant under the change $a \to \pm a$, $a_\infty \to \pm a_\infty$. Finally, the instanton partition function for real $t$ is a real analytic function of its parameters, $F^*(a, a_0, a_t, a_1, a_\infty) = F(a^*, a_0^*, a_t^*, a_1^*, a_\infty^*)$. The property (27) then follows.

For the black brane, upon taking the limit (20) the result takes the form

$$G_R^{\mathrm{brane}}(\omega, |\mathbf{k}|) = \pi^{4a_1} \frac{G_R(\omega, \ell)}{\left(1 + R_+^2\right)^{2a_1}}, \tag{28}$$

where $G_R(\omega, \ell)$ is taken from (26), but $a_i$, $t$, and $u$ are now mapped to $(\omega, \mathbf{k})$ according to Table II and equation (19). In (28) the temperature for the theory on $S^1 \times \mathbb{R}^3$ is set to 1.

The exact expressions presented above involve in a crucial way the NS free energy. As explained in Appendix C, the NS free energy is computed as a (convergent) series expansion in the instanton counting parameter $t$,

$$F = \sum_{n \geq 1}^{\infty} c_n(a, a_0, a_t, a_1, a_\infty) t^n . \tag{29}$$

The coefficients $c_n(a, a_0, a_t, a_1, a_\infty)$ in this series have a precise combinatorial definition in terms of Young diagrams. Hence in principle we can determine all of them. Given (29) one can straightforwardly solve the Matone relation (21) as a series in $t$ as well.

We can also write the above equation in a compact way by using the full NS free energy $F^{\mathrm{NS}}$ (C6), which is the sum of the instanton part $F$, the one-loop part $F^{1-\mathrm{loop}}$, and the classical term $F^{\mathrm{p}} = -2a \log t$. The formula becomes

$$G_R(\omega, \ell) = (1 + R_+^2)^{2a_1} \frac{\Gamma(-2a_1)}{\Gamma(2a_1)} \frac{\mathcal{G}(t, a, a_0, a_1, a_\infty, a_t)}{\mathcal{G}(t, a, a_0, -a_1, a_\infty, a_t)} \tag{30}$$

with

$$\mathcal{G}(t, a, a_0, a_1, a_\infty, a_t) = e^{-\frac{1}{2}\partial_{a_1}F^{\mathrm{NS}}} \sinh\left(\frac{\partial_a F^{\mathrm{NS}}}{2}\right) . \tag{31}$$

This is the typical form of the Fredholm determinant in this class of theories [80, eq. 8.12], [81, eq. 5.6], see also [82, 83]. Note that the result for the two-point function has the following simple property: under $\Delta \to 4 - \Delta$

we have $G_R \to \frac{1}{G_R}$. This property is manifest in (30) after noticing that under this transformation $a_1 \to -a_1$. It is also expected on general grounds because sending $\Delta \to 4 - \Delta$ switches the boundary conditions [84], so that the source and response are interchanged.

Let us conclude this section with a practical comment. When doing the actual computations we truncate the series in $t$ at some maximal instanton number $n_{\mathrm{max}}$. Given $n_{\mathrm{max}}$ and the corresponding $F^{n_{\mathrm{max}}}$, we then solve (21) for $a$ as a function of $u$ perturbatively in $t$. This step requires solving a linear equation at every new order in $t$. Finally, we plug both $F^{n_{\mathrm{max}}}$ and $a^{n_{\mathrm{max}}}(u)$ in (30) and evaluate $G_R^{n_{\mathrm{max}}}(\omega, \ell)$. We present an example of this procedure for $n_{\mathrm{max}} \leq 7$ and the case of the black brane in Figure 1.[6] We find a beautiful agreement between our result and the direct numerical solution of the wave equation.

With the methods we used, going to higher $n_{\mathrm{max}}$ gets computationally costly rather quickly. For example, in the case of the $N_f = 4$ theory that we are interested in, going beyond 5-10 instantons appears challenging on a laptop. Hence to fully exploit the power of our method it would be important to identify the range of parameters for which $G_R(\omega, \ell)$ can be reliably computed with a few instantons. It would also be desirable to develop a more efficient way of computing the NS functions (either analytically or numerically).[7]

## IV. RELATION TO THE HEAVY-LIGHT CONFORMAL BOOTSTRAP

The thermal two-point function computed in the previous section is directly related to the four-point correlation function of local operators $\langle \mathcal{O}_H \mathcal{O}_L \mathcal{O}_L \mathcal{O}_H \rangle$ [86, 87]. Here $\mathcal{O}_L$ is the light or probe operator of dimension $\Delta_L$ from the previous section,[8] and $\mathcal{O}_H$ is a heavy operator with $\Delta_H \sim c_T$ that is dual to a black hole microstate, where $c_T$ parameterizes the two-point function of canonically normalized stress tensors. For the precise relationship between $\mu \sim \frac{\Delta_H}{c_T}$, $\Delta_H$ and $c_T$ see e.g. [48].

More precisely, we define the four-point function as follows

$$G(z, \bar{z}) \equiv \langle \mathcal{O}_H(0) \mathcal{O}_L(z, \bar{z}) \mathcal{O}_L(1, 1) \mathcal{O}_H(\infty) \rangle, \tag{32}$$

where all operators for simplicity are taken to be real scalars. The insertion at infinity is given by $\mathcal{O}_H(\infty) =$

---

[6] Alternatively, we can use (26) to compute $G_R(\omega, a)$ and we can use (21) to evaluate the map $\ell(\omega, a)$ (or $\mathbf{k}(\omega, a)$). This is possible because the dependence on spin $\ell$ (or momentum $\mathbf{k}$) enters the problem only through the parameter $u$, which does not appear in the exact formula (26).

[7] For example using TBA-like techniques as in [85] and references there.

[8] In this section we switch from $\Delta$ to $\Delta_L$ to make the distinction between the light and heavy operators more obvious.

$\lim_{x_4 \to \infty} |x_4|^{2\Delta_H} \mathcal{O}_H(x_4)$. We also used conformal symmetry to put all four operators in a two-dimensional plane with coordinate $z = x^1 + ix^2$.

We choose the normalization of operators such that in the short distance limit $z, \bar{z} \to 1$ we have

$$G(z, \bar{z}) = \frac{1}{(1-z)^{\Delta_L}(1-\bar{z})^{\Delta_L}} + \dots . \qquad (33)$$

This four-point function admits an OPE expansion in various channels, see e.g. [88]. We focus on the heavy-light channel, in which the expansion of the four-point function takes the form

$$G(z, \bar{z}) = \sum_{\mathcal{O}_{\Delta,\ell}} \lambda^2_{H,L,\mathcal{O}_{\Delta,\ell}} \frac{g^{\Delta_{H,L},-\Delta_{H,L}}_{\Delta,\ell}(z, \bar{z})}{(z\bar{z})^{\frac{1}{2}(\Delta_H+\Delta_L)}}, \qquad (34)$$

where $\Delta_{H,L} \equiv \Delta_H - \Delta_L$, and $\lambda_{H,L,\mathcal{O}_{\Delta,\ell}} \in \mathbb{R}$ are the three-point functions. Finally, the expressions for the conformal blocks $g^{\Delta_{H,L},-\Delta_{H,L}}_{\Delta,\ell}(z, \bar{z})$ can be found for example in [89, 90].

We next consider the $\Delta_H, c_T \to \infty$ limit of the expansion of $G(z, \bar{z})$ above with $\mu = \frac{160}{3}\frac{\Delta_H}{c_T}$ kept fixed. In this limit the spectrum of operators becomes effectively continuous and the contribution of descendants is suppressed [88].[9] Specializing to $d = 4$, we get the following expression for the OPE expansion,

$$G(z, \bar{z}) = \sum_{\ell=0}^{\infty} \int_{-\infty}^{\infty} d\omega \, g_{\omega,\ell}(z\bar{z})^{\frac{\omega-\Delta-\ell}{2}} \frac{z^{\ell+1} - \bar{z}^{\ell+1}}{z - \bar{z}} , \qquad (35)$$

where we introduced $\omega = \Delta'_H - \Delta_H$, and $g_{\omega,\ell}$ for the product of the three-point functions $\lambda^2_{H,L,\mathcal{O}_{\Delta_{H'},\ell}}$ and the density of primaries. Thanks to unitarity we have $g_{\omega,\ell} \geq 0$ and KMS symmetry implies that

$$g_{-\omega,\ell} = e^{-\beta\omega} g_{\omega,\ell} . \qquad (36)$$

We can now state the precise relationship between the heavy-light four-point function and the thermal two-point function [26],

$$g_{\omega,\ell} = \frac{\ell+1}{2\pi(\Delta_L-1)(\Delta_L-2)} \frac{\operatorname{Im} G_R(\omega,\ell)}{1-e^{-\beta\omega}} , \qquad (37)$$

where $\beta$ and $\Delta_H$ are related in the standard way, $\beta = \frac{\partial S(\Delta_H)}{\partial \Delta_H}$. In this formula $S(\Delta_H)$ is the effective density of primaries of dimension $\Delta_H$. This relation is the combination of the eigenstate thermalization hypothesis [86, 87, 91, 92] and the standard relations between various thermal two-point functions [79]. The factor $\ell+1$ originates from summing over $\vec{m}$ of the spherical harmonics $Y_{\ell\vec{m}}$, see Appendix A of [21] for details.

There is a natural limit in which the general expression (37) simplifies: it is the large spin limit $\ell \to \infty$. As explained in detail in [25, 26], in this limit the relevant states are orbits which are stable perturbatively in $\frac{1}{\ell}$. These states manifest themselves in $G_R(\omega,\ell)$ as poles (also known as quasi-normal modes) with imaginary part which is non-perturbative in spin $\ell$. Therefore, perturbatively in $\ell$, $\operatorname{Im} G_R(\omega,\ell)$ effectively becomes the sum of $\delta(|\omega| - \omega_{n\ell})$, where $\omega_{n\ell} = \Delta_L + \ell + 2n + \gamma_{n\ell}$ and $\gamma_{n\ell} \to 0$ at large spin. Notice that for $|\omega| \sim \ell$, $[(1-e^{-\beta\omega})^{-1}]_{\text{pert}}$ becomes a step function $\theta(\omega)$, and in this way $g_{\omega,\ell}$ reduces at large spin to the expected sum over heavy-light double-twist operators $\mathcal{O}_H \square^n \partial^\ell \mathcal{O}_L$ .

We can summarize this as follows

$$g^{\text{pert}}_{\omega,\ell} = \theta(\omega) \frac{\ell+1}{2\pi(\Delta_L-1)(\Delta_L-2)} \operatorname{Im} G^{\text{pert}}_R(\omega,\ell)$$
$$= \sum_{n=0}^{\infty} c_{n\ell} \delta(\omega - \omega_{n\ell}) , \qquad (38)$$

where the relation holds for all the terms which contribute as powers at large spin $\ell$, namely $\frac{1}{\ell^{\#}}$. We signified this by writing $\operatorname{Im} G^{\text{pert}}_R(\omega,\ell)$ (see also Section V for a more precise definition). Here $c_{n\ell}$ is the square of the OPE coefficients of double-twist operators. In writing (38) we also used the fact that at fixed $\omega$, $\operatorname{Im} G_R(\omega,\ell)$ is nonperturbative in spin at large $\ell$.[10] We establish this fact in Appendix F.

The large spin expansion of the heavy-light four-point function was actively explored in the last few years [48–61]. One of the basic observations of these works is that in $d > 2$ the effective expansion parameter is $\frac{\mu}{\ell^{\frac{d-2}{2}}}$. We can therefore equivalently study the small $\mu$ expansion of the exact results. This is what we do in the next section.

## V. SMALL $\mu$ EXPANSION

In the previous section we explained how to compute the dimensions and OPE data of heavy-light double-twist operators using the exact two-point function (26). Now we would like to carry out this procedure perturbatively in $1/\ell$. Note that the expected perturbative parameter is $\frac{\mu}{\ell}$ [48–61], so that instead of taking the large spin limit, we can equivalently consider the limit of small black holes. This is a natural limit from the point of view of the Nekrasov-Shatashvili functions, which are defined as a perturbative expansion in $t \sim \mu$ for small $\mu$.

_______

[9]This requires an extra assumption on which operators dominate the OPE, see e.g. the discussion in [26].

_______

[10]In principle, non-perturbative in spin effects are accessible to the light-cone bootstrap [93] thanks to the Lorentzian inversion formula [94–96]. However, such effects have not been yet explored in the context of the heavy-light bootstrap.

## A. Exact quantization condition and residues

In the small $\mu$ and large spin expansion, the Green's function (26) simplifies considerably. To see this, note that at small $\mu$ the Matone relation (21) becomes

$$a = \pm \frac{\ell + 1}{2} + \mathcal{O}(\mu), \qquad (39)$$

where we plugged in the dictionary from Table I. Since the Green's function is invariant under $a \to -a$, it does not matter what sign we pick in (39). Choosing the minus sign in (39), the ratio of the $\sigma = -1$ term to the $\sigma = 1$ term in both the numerator and the denominator of (26) scales as $\mu^{\ell+1}$, which is exponentially small in spin. Neglecting this nonperturbative correction, we find

$$G_R^{\text{pert}}(\omega, \ell) = (1 + R_+^2)^{2a_1} e^{-\partial_{a_1} F}$$
$$\frac{\Gamma(-2a_1)\Gamma(1/2 - a + a_1 - a_\infty)\Gamma(1/2 - a + a_1 + a_\infty)}{\Gamma(2a_1)\Gamma(1/2 - a - a_1 - a_\infty)\Gamma(1/2 - a - a_1 + a_\infty)}. \qquad (40)$$

In a sense, this expression is a generalization of the semi-classical Virasoro vacuum block [97, 98] to $d = 4$. Indeed, via (38) it encodes the contribution of the identity and multi-stress tensor contributions in the light-light channel, schematically $\mathcal{O}_L \times \mathcal{O}_L \sim 1 + T + T^2 + ...$. The effects non-perturbative in spin (which are intimately related to the presence of the black hole horizon) are, on the other hand, encoded in the contribution of the double-twist operators $\mathcal{O}_L \times \mathcal{O}_L \sim \mathcal{O}_L \square^n \partial^\ell \mathcal{O}_L$.

We can now explicitly read off the poles and residues of (40). There are poles in the function $\Gamma(1/2 - a + a_1 - a_\infty)$ at positive energies $\omega = \omega_{n\ell}$, which are nothing but the dimensions of the double-twist operators. The locations of these poles are determined by the following quantization condition,

$$\omega_{n\ell}: \quad n = a + a_\infty - a_1 - 1/2, \quad n \geq 0. \qquad (41)$$

Geometrically this corresponds to the quantization of the quantum A-period associated to the Seiberg-Witten geometry. The relation (41) implicitly defines the scaling dimensions of the double-twist operators $\omega_{n,\ell}$ via the black hole to gauge theory dictionary in Table I and (15), along with the Matone relation (21). Computing the residues of the two-point function (40) and using (38) and Table II then gives

$$c_{n\ell} = \frac{(\ell + 1)\Gamma(\Delta + n - 1)\Gamma(2a_\infty - n)}{\Gamma(\Delta)\Gamma(\Delta - 1)\Gamma(n + 1)\Gamma(2a_\infty - n - \Delta + 2)}$$
$$\times \frac{(1 + R_+^2)^{\Delta - 2} e^{-\partial_{a_1} F}}{2} \left( \frac{d(a + a_\infty)}{d\omega} \right)^{-1} \Big|_{\omega = \omega_{n\ell}}. \qquad (42)$$

Note that, since $F$ is defined by a power series in $\mu$ whose coefficients are rational functions, it is straightforward to invert (41) to any desired order in $\mu$ by perturbing around the $\mu = 0$ result. In this sense, (41) and (42) represent an exact solution for the bootstrap data.

## B. Anomalous dimensions and OPE data

To organize the perturbative series, let us define

$$\omega_{n\ell} = \omega_{n\ell}^{(0)} + \sum_{i=1}^{\infty} \mu^i \gamma_{n\ell}^{(i)} ,$$
$$c_{n\ell} = c_{n\ell}^{(0)} \left( 1 + \sum_{i=1}^{\infty} \mu^i c_{n\ell}^{(i)} \right) . \qquad (43)$$

We then plug these expansions into (41) and (40), using the dictionary in Table I and (15), the Matone relation (21), and the definitions in Appendix C. At zeroth order in $\mu$, we reproduce the OPE coefficients in generalized free field theory, see e.g. [49, 59],

$$\omega_{n\ell}^{(0)} = \Delta + \ell + 2n , \qquad (44)$$
$$c_{n\ell}^{(0)} = \frac{(\ell + 1)\Gamma(\Delta + n - 1)\Gamma(\Delta + n + \ell)}{\Gamma(\Delta)\Gamma(\Delta - 1)\Gamma(n + 1)\Gamma(n + \ell + 2)}, \qquad (45)$$

namely we have the following identity

$$\sum_{n,\ell=0}^{\infty} c_{n\ell}^{(0)} (z\bar{z})^{\frac{\omega_{n\ell}^{(0)} - \Delta - \ell}{2}} \frac{z^{\ell+1} - \bar{z}^{\ell+1}}{z - \bar{z}} = \frac{1}{(1 - z)^\Delta (1 - \bar{z})^\Delta} . \qquad (46)$$

Now let us go to first order in $\mu$. We find

$$\gamma_{n\ell}^{(1)} = -\frac{\Delta^2 + \Delta(6n - 1) + 6n(n - 1)}{2(\ell + 1)} , \qquad (47)$$
$$c_{n\ell}^{(1)} = \frac{1}{2} \left( 3(\Delta - 2) - \frac{3(\Delta + 2n - 1)}{\ell + 1} + \qquad (48)$$
$$(3(\ell + 2n + \Delta) - 2\gamma_1)(\psi^{(0)}(2 + \ell + n) - \psi^{(0)}(\Delta + \ell + n)) \right),$$

where $\psi^{(m)}(x) = d^{m+1} \log \Gamma(x)/dx^{m+1}$ is the polygamma function of order $m$. These results agree with the light-cone bootstrap computations [52, 59, 61].

At second order $\mathcal{O}(\mu^2)$ the answers become more complicated, and are displayed explicitly in Appendix D. Already at this order only $\mathcal{O}(1/\ell^2)$ results are available in the literature, which is the leading term in the large spin expansion. We find complete agreement with the result of [59].

At $k$-th order $\mathcal{O}(\mu^k)$ we find the following structure

$$\gamma_{n\ell}^{(k)} = \sum_{j=0}^{2k+1} R_j^{(k)}(n, \ell)\Delta^j, \qquad (49)$$

where $R_j^{(k)}(n, \ell)$ are polynomials of degree $k - j$ in $n$ and are meromorphic functions of $\ell$. The singularities occur at $\ell_{\text{sing}} \in \mathbb{Z}$ and $-k - 1 \leq \ell_{\text{sing}} \leq k - 1$. These singularities are however spurious and occur because for $\ell < k$ it is not justified to drop the $\sigma = -1$ term when going from (26) to (40).

For the three-point functions $c_{n\ell}^{(k)}$ the structure is very

similar, the main difference being that the analogs of $R_j^{(k)}(n, \ell)$ can also depend on $\psi^{(m)}(\Delta + n + \ell) - \psi^{(m)}(2 + n + \ell)$ with $m \leq k - 1$.

We refer the interested reader to the text files attached to the submission for the full expressions of $\gamma_{n\ell}^{(k)}$ (`gammas.txt`) and $c_{n\ell}^{(k)}$ (`cs.txt`) to order $k \leq 5$.

### C. The imaginary part of quasi-normal modes

Until now, in computing the position of the poles of $G_R(\omega, \ell)$, we have neglected the imaginary part, which is exponentially suppressed at large spin.[11] Let us now compute the leading behavior of the imaginary part, for which we must consider the exact Green's function (26). In the large spin expansion, the numerator of (26) is finite, so the poles arise when the denominator vanishes. Therefore we must solve

$$0 = \sum_{\sigma = \pm} \mathcal{M}_{-\sigma}(a_t, a; a_0)\mathcal{M}_{(-\sigma)-}(a, a_1, a_\infty)t^{\sigma a}e^{-\frac{\sigma}{2}\partial_a F}. \tag{50}$$

We make an ansatz

$$\text{Im } \omega_{n\ell} = i \sum_{k=1}^{\infty} f_{n\ell}^{(k)} \mu^{\ell+1/2+k}, \tag{51}$$

where $f_{n\ell}^{(k)}$ are real. Note that the imaginary part behaves as $\mu^\ell$ at large $\ell$, as expected from the tunneling calculation in [26]. The first contribution to the imaginary part is at order $\mu^{\ell+3/2}$, which is consistent with numerical evidence [99]. As shown in Appendix E, the explicit form of the leading contribution to the imaginary part is

$$f_{n\ell}^{(1)} = -\frac{2^{-4\ell}\pi^2}{(\ell+1)^2}\omega_{n\ell}^{(0)}\frac{\Gamma(\Delta+n+\ell)}{\Gamma(\Delta+n-1)}\frac{\Gamma(n+\ell+2)}{\Gamma(n+1)\Gamma(\frac{\ell+1}{2})^4}. \tag{52}$$

It should be possible to check this expression using the techniques of [100]. Note that $\text{Im } \omega_{n\ell} < 0$ as expected from causality.

## VI. CONCLUSIONS AND FUTURE DIRECTIONS

In this paper we have computed the holographic thermal scalar two-point function $\langle \mathcal{O}\mathcal{O}\rangle_\beta$. Via the AdS/CFT correspondence, the problem reduces to the study of wave propagation on the AdS-Schwarzschild background. To solve the problem we used the connection between the

wave equation on the AdS-Schwarzschild background and four-dimensional supersymmetric gauge theories. The result for the two-point function for a four-dimensional holographic CFT on $S^1 \times S^3$ dual to a black hole geometry is the formula (26). For a holographic CFT on $S^1 \times \mathbb{R}^3$ dual to a black brane geometry the result is (28). A key ingredient that enters into our formulae is the Nekrasov-Shatashvili instanton partition function $F$ of an $SU(2)$ supersymmetric gauge theory with four fundamental hypermultiplets.

We analyzed the exact formulas analytically in three different regimes:

- Large $\omega$, fixed spin $\ell$/momentum $\mathbf{k}$ limit. This limit is controlled by the OPE between the probe operators, $\mathcal{O} \times \mathcal{O} = 1 + ...$ [101, 102]. To leading order the result is identical for the black hole and black brane and takes the form (B13). Although we were not able to obtain a complete analytic understanding of this limit, we showed that the exact result reduces to (B13) up to an overall constant. We then checked using the instanton expansion that this constant approaches 1 as we increase the instanton number.

- Fixed $\omega$, large spin $\ell$/large momentum $\mathbf{k}$ limit. As reviewed in Appendix F, the NS partition function can be computed exactly in this limit [103]. For the black hole the large spin asymptotic behavior is given by (F3) and (F5), and for the black brane the result is given by (F6). Our results for the imaginary part of the black brane two-point function agree with those previously computed in [29].

- Small $\frac{\mu}{\ell}$/light-cone bootstrap expansion. In this limit the spectrum becomes effectively discrete and our formulas reduce to a sum over double-twist heavy-light operators. The all-order solution to the light-cone bootstrap is encoded in formula (40). At order $\mathcal{O}(\mu^1)$ we reproduced the known results. At order $\mathcal{O}(\mu^2)$ only leading at large spin data is publicly available and agrees with our results. As a supplement to our submission we provide explicit formulas for the double-twist data up to order $\mathcal{O}(\mu^5)$. We also derived the leading small $\mu$ asymptotic behavior of the imaginary part of quasi-normal modes, see (47).

We also analyzed the exact formulas numerically by truncating the instanton sum (29) to some finite value $n_{\max}$. In this work we only limited ourselves to $n_{\max} \leq 7$. An example of a result that cannot be derived by any known analytical methods is shown in Figure 1. We have found an excellent agreement with the direct numerical solution of the differential equation.

Our paper only embarks upon an exploration of a fascinating connection between finite-temperature correlators and supersymmetric gauge theories. There are many future directions to explore and we end our paper with naming an obvious few.

---

[11] Physically, this is related to the fact that classically stable orbits can decay quantum-mechanically due to tunneling, see e.g. [24].

- In this paper we have restricted our analysis to $d = 4$ and a black hole with zero charge and spin. It would be very interesting to generalize our analysis to general $d$, and to consider spinning and charged black holes, as well as spinning and charged probes. In the latter case, considering the two-point function of conserved currents $\langle J_\mu J_\nu \rangle_\beta$ and stress-energy tensors $\langle T_{\mu\nu} T_{\rho\sigma} \rangle_\beta$ is particularly interesting due to their relation to transport and hydrodynamics, see e.g. [104–106]. The corresponding stress-tensor OPE expansion was analyzed in [107].

- Another obvious extension is to consider thermal higher-point functions, e.g. the out-of-time-ordered four-point function [18], as well as to study gravitational loop effects for the two-point function [108]. In the bulk such computations correspond to going beyond linear order, and they require knowledge of the bulk-to-boundary and bulk-to-bulk propagators on the black hole background. In the language of [36] these are given in terms of the Virasoro conformal blocks and via the AGT correspondence can be again expressed in terms of the instanton partition functions.

- To make the exact formulas particularly useful it is important to develop efficient numerical algorithms to evaluate them approximately. The most obvious approach is to truncate the number of instantons at some value $n_{\max}$. This is the approach that we adopted in this paper and we obtained accurate results, see Figure 1. We have not systematically explored the space of parameters that can be effectively probed using the truncated instanton number sum and we leave this for the future work.

- From the point of view of conformal bootstrap our results concern the heavy-heavy-light-light four-point function viewed from the heavy-light channel, see Section IV. In the same sense the all-order formula (40) solves the light-cone bootstrap in the heavy-light channel. Intriguing structures have been recently observed in the light-light channel [51, 53, 56], which is related to our work by crossing. It would be very interesting to bridge the results of our work and these recent developments.

- At zero temperatures there is a simple correspondence between perturbative solutions to crossing equations and effective field theories in AdS [1]. A similar connection was explored in [27] in the thermal AdS phase, thanks to the fact that the relevant "unperturbed" finite temperature generalized free field solution is explicitly known, see e.g. [109]. An exciting problem in this context is to understand a similar connection between crossing and effective field theories in AdS in the black hole phase. Here our exact formula provides an unperturbed seed solution, around which perturbations can be studied. It would be very interesting to explore this possibility and more generally explore consistency of holographic conformal field theories at finite temperatures.

- In [20, 21] the two-point function was used to analyze subtle signatures of the black hole singularity. It would be very interesting to revisit this problem given the exact two-point function, and to identify what the black hole interior corresponds to on the instanton partition function side.

## ACKNOWLEDGMENTS

We thank Paolo Arnaudo, Giulio Bonelli, Shouvik Datta, Shota Komatsu, Yue-Zhou Li, Baur Mukhametzhanov, Kyriakos Papadodimas, Gábor Sárosi, Wilke van der Schee and Alessandro Tanzini for useful discussions. We also thank the organizers of the hybrid GGI workshop "New horizons for (no-)horizon physics: from gauge to gravity and back" as well as the organizers of the in-person Eurostrings 2022 conference, where some of us got to know each other and started collaborating. This project has received funding from the European Research Council (ERC) under the European Union's Horizon 2020 research and innovation programme (grant agreement number 949077). The work of AG is partially supported by the Fonds National Suisse, Grant No. 185723 and by the NCCR "The Mathematics of Physics" (SwissMAP).

## Appendix A: Conventions

Here we collect our conventions for various thermal two-point functions. Let us start with the case of the black hole. This is dual to a holographic CFT on $S^1 \times S^3$, with the radius of $S^1$ being $\beta$ and the radius of $S^3$ set to 1. We have for the retarded two-point function

$$i\theta(t)\langle[\mathcal{O}(t,\vec{n}), \mathcal{O}(0,\vec{n}')]\rangle_\beta = \frac{1}{4\pi(\Delta-1)(\Delta-2)} \int_{-\infty}^{\infty} d\omega \, e^{-i\omega t} \sum_{\ell=0}^{\infty} (\ell+1) G_R(\omega,\ell) \frac{\sin(\ell+1)\theta}{\sin\theta}, \qquad (A1)$$

where $\vec{n} \cdot \vec{n}' = \cos\theta$ and $\vec{n}^2 = \vec{n}'^2 = 1$, so that $\vec{n}, \vec{n}' \in S^3$. $G_R(\omega,\ell)$ is given by (26). We also used for partial waves $C_\ell^{(1)}(\cos\theta) = \frac{\sin(\ell+1)\theta}{\sin\theta}$.

For the Euclidean two-point function we have

$$\langle \mathcal{O}(\tau, \vec{n}) \mathcal{O}(0, \vec{n}') \rangle_\beta = \int_{-\infty}^{\infty} d\omega\, e^{-\omega\tau} \sum_{\ell=0}^{\infty} g_{\omega,\ell} \frac{\sin(\ell+1)\theta}{\sin\theta}, \quad 0 < \tau < \beta, \tag{A2}$$

where $g_{\omega,\ell}$ is given in (37) and $\tau$ is the Euclidean time. KMS symmetry or invariance under $\tau \to \beta - \tau$ holds thanks to (36). We normalize the operators such that the unit operator contributes as $\frac{e^{-\tau\Delta}}{([1-e^{-\tau+i\theta}][1-e^{-\tau-i\theta}])^\Delta}$. The Wightman function can be obtained through Wick rotation by taking $\tau \to \epsilon + it$ and then $\epsilon \to 0$.

For the black brane, or holographic CFT on $S^1 \times \mathbb{R}^{d-1}$ with the radius of $S^1$ set to 1, we have for the retarded two-point function

$$i\theta(t)\langle[\mathcal{O}(t,\mathbf{x}), \mathcal{O}(0,0)]\rangle_{\beta=1} = \frac{1}{(4\pi)^2(\Delta-1)(\Delta-2)} \int_{-\infty}^{\infty} d\omega\, e^{-i\omega t} \int_{-\infty}^{\infty} d^3\mathbf{k}\, e^{i\mathbf{k}\cdot\mathbf{x}} G_R^{\text{brane}}(\omega, \mathbf{k}). \tag{A3}$$

$G_R^{\text{brane}}(\omega, \mathbf{k})$ is given by (28).

For the Euclidean two-point function we have

$$\langle \mathcal{O}(\tau, \mathbf{x}) \mathcal{O}(0, 0) \rangle_{\beta=1} = \frac{1}{4\pi} \int_{-\infty}^{\infty} d\omega\, e^{-\omega\tau} \int_{-\infty}^{\infty} d^3\mathbf{k}\, e^{i\mathbf{k}\cdot\mathbf{x}} g_{\omega,\mathbf{k}}, \quad 0 < \tau < 1, \tag{A4}$$

where $\tau$ is the Euclidean time and $g_{\omega,\mathbf{k}}$ is given by (B2). We normalize operators such that the unit operator contributes as $\frac{1}{(\tau^2+\mathbf{x}^2)^\Delta}$. KMS symmetry or invariance under $\tau \to 1 - \tau$ holds thanks to (B5). The Wightman function can be obtained through Wick rotation by taking $\tau \to \epsilon + it$ and then $\epsilon \to 0$.

## Appendix B: From black hole to black brane

Let us describe in a bit more detail the infinite temperature limit that takes us from the black hole to the black brane. This is one example of the so-called macroscopic limits considered in [88] and we simply apply the formulas of that paper to our case.

First of all, we introduce the limiting retarded two-point function as follows,

$$G_R^{\text{brane}}(\omega, |\mathbf{k}|) = \lim_{T\to\infty} \frac{G_R(\omega T, |\mathbf{k}|T)}{T^{4a_1}}, \tag{B1}$$

where $G_R(\omega, |\mathbf{k}|)$ is the retarded thermal two-point function for a CFT on $S^1 \times \mathbb{R}^3$ with $(\omega, |\mathbf{k}|)$ measured in units of temperature on $S^1$. Let us also introduce

$$g_{\omega,\mathbf{k}}^{\text{brane}} = \frac{1}{2\pi(\Delta-1)(\Delta-2)} \frac{\text{Im}\, G_R^{\text{brane}}(\omega, |\mathbf{k}|)}{1 - e^{-\omega}}. \tag{B2}$$

At the level of the two-point function we consider the following limit

$$G^{\text{brane}}(w, \bar{w}) \equiv \lim_{T\to\infty} T^{-2\Delta} G\left(z = 1 - \frac{w}{T}, \bar{z} = 1 - \frac{\bar{w}}{T}\right). \tag{B3}$$

Plugging this formula in the OPE expansion (35) we get

$$G^{\text{brane}}(w, \bar{w}) = \lim_{T\to\infty} T^{-4} \int_0^\infty d|\mathbf{k}||\mathbf{k}| \times T^2 \int_{-\infty}^\infty d\omega \times T\, g_{\omega,\mathbf{k}} e^{-\frac{(w+\bar{w})}{2}(\omega-|\mathbf{k}|)} \frac{e^{-w|\mathbf{k}|} - e^{-\bar{w}|\mathbf{k}|}}{\bar{w} - w} \times T$$

$$= \int_0^\infty d|\mathbf{k}||\mathbf{k}| \int_{-\infty}^\infty d\omega\, g_{\omega,\mathbf{k}} e^{-\frac{(w+\bar{w})}{2}(\omega-|\mathbf{k}|)} \frac{e^{-w|\mathbf{k}|} - e^{-\bar{w}|\mathbf{k}|}}{\bar{w} - w}, \tag{B4}$$

where we converted the sum to an integral, $\sum_\ell \to T \int d|\mathbf{k}|$.

The KMS symmetry becomes

$$g_{-\omega,\mathbf{k}} = e^{-\omega} g_{\omega,\mathbf{k}}. \tag{B5}$$

We next consider the two-point function on $S^1 \times \mathbb{R}^{d-1}$,

$$\langle \mathcal{O}(\tau, \mathbf{x}) \mathcal{O}(0,0) \rangle_\beta = G^{\text{brane}} \Big( \tau + i|\mathbf{x}|, \tau - i|\mathbf{x}| \Big). \tag{B6}$$

In terms of these variables we get

$$\langle \mathcal{O}(\tau, \mathbf{x}) \mathcal{O}(0,0) \rangle_\beta = \int_0^\infty d|\mathbf{k}| \, |\mathbf{k}| \int_{-\infty}^\infty d\omega \, g_{\omega, \mathbf{k}} e^{-\omega \tau} \frac{\sin |\mathbf{k}| |\mathbf{x}|}{|\mathbf{x}|}$$

$$= \frac{1}{4\pi} \int_{-\infty}^\infty d^3\mathbf{k} \int_{-\infty}^\infty d\omega \, e^{i\mathbf{k} \cdot \mathbf{x}} e^{-\omega \tau} g_{\omega, \mathbf{k}}. \tag{B7}$$

The result is indeed invariant under KMS symmetry $\tau \to 1 - \tau$ (recall that we have set $\beta = 1$). By analytically continuing to Lorentzian time we see that $g_{\omega, \mathbf{k}}$ is the Fourier transform of the Wightman two-point function.

Note that taking the limit (B3) does not change the normalization of the scalar operator, since

$$\lim_{T \to \infty} T^{-2\Delta} \frac{1}{\left(1 - (1 - \frac{w}{T})\right)^\Delta \left(1 - (1 - \frac{\bar{w}}{T})\right)^\Delta} = \frac{1}{(w\bar{w})^\Delta} = \frac{1}{(\tau^2 + \mathbf{x}^2)^\Delta}. \tag{B8}$$

In other words if the operator was unit-normalized it will continue to be unit-normalized after taking the limit.

Let us finish with a few formulas for the vacuum correlators. In Fourier space, the vacuum Wightman two-point function $\langle \mathcal{O}(t, \mathbf{x}) \mathcal{O}(0,0) \rangle_0 = \frac{1}{(-(t-i\epsilon)^2 + \mathbf{x}^2)^\Delta}$ takes the form

$$\int_{-\infty}^\infty dt \, d^3\mathbf{x} \, e^{i\omega t - i\mathbf{k} \cdot \mathbf{x}} \frac{1}{(-(t - i\epsilon)^2 + \mathbf{x}^2)^\Delta} = \theta(\omega)\theta(\omega^2 - \mathbf{k}^2) \frac{2\pi^3}{\Gamma(\Delta)\Gamma(\Delta - 1)} \left( \frac{\omega^2 - \mathbf{k}^2}{4} \right)^{\Delta - 2}. \tag{B9}$$

It is expected that (B9) controls the large $\omega$ asymptotics of the thermal correlators [101, 102].

From (B7) we get that

$$g_{\omega, \mathbf{k}} = \lim_{\epsilon \to 0} \frac{1}{4\pi^3} \int_{-\infty}^\infty d^3\mathbf{k} \int_{-\infty}^\infty d\omega \, e^{-i\mathbf{k} \cdot \mathbf{x}} e^{it\omega} \langle \mathcal{O}(\epsilon + it, \mathbf{x}) \mathcal{O}(0,0) \rangle_\beta. \tag{B10}$$

Formulas (B9), (B5) together with (B2) imply that

$$\lim_{|\omega| \gg 1, |\omega| \gg |\mathbf{k}|} \text{Im} \, G_R^{\text{brane}}(\omega, |\mathbf{k}|) \simeq -\sin \pi \Delta \frac{\Gamma(2 - \Delta)}{\Gamma(\Delta - 2)} \text{sign}(\omega) \left( \frac{|\omega|}{2} \right)^{2(\Delta - 2)}. \tag{B11}$$

Via dispersion relations for $G_R^{\text{brane}}(\omega, |\mathbf{k}|)$ this leads to the following asymptotic behavior for the real part,

$$\lim_{|\omega| \gg 1, |\omega| \gg |\mathbf{k}|} \text{Re} \, G_R^{\text{brane}}(\omega, |\mathbf{k}|) \simeq \cos \pi \Delta \frac{\Gamma(2 - \Delta)}{\Gamma(\Delta - 2)} \left( \frac{|\omega|}{2} \right)^{2(\Delta - 2)}, \tag{B12}$$

where everywhere we tacitly assumed that $\Delta$ is not an integer. For the black hole case ($t < \frac{1}{2}$) we get in the same way

$$\lim_{|\omega|/T \gg 1, \ell} G_R(\omega, \ell) \simeq e^{-\pi i \Delta \text{sign}(\omega)} \frac{\Gamma(2 - \Delta)}{\Gamma(\Delta - 2)} \left( \frac{|\omega|}{2} \right)^{2(\Delta - 2)}. \tag{B13}$$

We can also derive the large $\omega$ and fixed $\ell$ behavior of the Green's function directly from our exact expression (26). Let us start with the black hole case. By solving the Matone relation (21) order by order in the instanton expansion, one finds in this limit $\partial_a F = ic_1(t)\omega + \mathcal{O}(\omega^0)$, $\partial_{a_1} F = c_3(t)(\Delta - 2) + \mathcal{O}(\omega^{-1})$, and $a = ic_2(t)\omega + \mathcal{O}(\omega^0)$, with $c_i(t) \in \mathbb{R}$. Since the Green's function (26) is invariant under $a \to -a$, we can choose $c_2 > 0$ without loss of generality. With this specification, the $\sigma = 1$ term in (26) dominates over the $\sigma = -1$ term. Expanding the gamma functions at large $\omega$ and using the dictionary in Table I, we find

$$G_R(\omega, \ell) \approx (1 + R_+^2)^{2a_1} e^{-\partial_{a_1} F} \frac{\Gamma(-2a_1)}{\Gamma(2a_1)} (a_\infty - a)^{2a_1} (-a - a_\infty)^{2a_1}$$

$$\approx \frac{\Gamma(2 - \Delta)}{\Gamma(\Delta - 2)} \left( \frac{|\omega|}{2} \right)^{2(\Delta - 2)} e^{-\pi i \Delta \text{sign}(\omega)} \Big( c(t) \Big)^{\Delta - 2}, \tag{B14}$$

where

$$c(t) = \frac{e^{-c_3(t)}(1-t)(4c_2(t)^2 + 2t^2 - 3t + 1)}{1-2t}. \tag{B15}$$

The OPE predicts that $c(t) = 1$.

We do not have complete analytic control over the constants $c_2(t)$ and $c_3(t)$, but we checked that (B15) approaches 1 by computing the first few orders in the instanton expansion, see Figure 2. Hence we recover (B13). The black brane results (B11) and (B12) correspond to $t \to \frac{1}{2}$ in Figure 2.

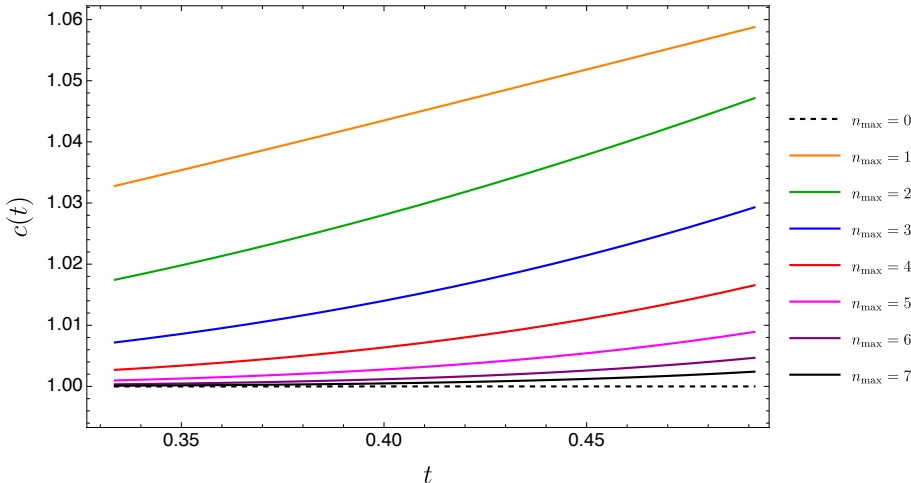

FIG. 2. $c(t)$ defined in (B15) as a function of the black hole mass (here parameterized by $t$), and the maximum instanton number $n_{\max}$. Based on the OPE we expect that $c(t)$ is independent of $t$ and is equal to 1.

## Appendix C: The Nekrasov-Shatashvili function

We denote by $Y = (\nu_1, \nu_2, ...)$ a partition (or Young tableau) and by $Y^t = (\nu_1^t, \nu_2^t, ...)$ its transpose. We also use $\vec{Y} = (Y_1, Y_2)$ to denote a vector of Young tableaux. The leg-length and the arm-length are defined by $h_Y(s) = \nu_j - i$ and $v_Y(s) = \nu_i^t - j$, where $s = (i, j)$ is a box. We define

$$z_{\rm h}\left(\vec{a}, \vec{Y}, \mu\right) = \prod_{I=1,2} \prod_{s \in Y_I} \left(a_I + \mu + \epsilon_1\left(i - \frac{1}{2}\right) + \epsilon_2\left(j - \frac{1}{2}\right)\right),$$

$$z_{\rm v}\left(\vec{a}, \vec{Y}\right) = \prod_{I,J=1}^{2} \prod_{s \in Y_I} \frac{1}{a_I - a_J - \epsilon_1 v_{Y_J}(s) + \epsilon_2(h_{Y_I}(s) + 1)} \prod_{s \in Y_J} \frac{1}{a_I - a_J + \epsilon_1(v_{Y_I}(s) + 1) - \epsilon_2 h_{Y_I}(s)}. \tag{C1}$$

In this paper we always take $\vec{a} = (a, -a)$ and $\epsilon_1 = 1$. The instanton part of the NS function is defined as

$$F = \lim_{\epsilon_2 \to 0} \epsilon_2 \log\left((1-t)^{-2\epsilon_2^{-1}(\frac{1}{2}+a_1)(\frac{1}{2}+a_t)} \sum_{\vec{Y}} t^{|\vec{Y}|} z_{\rm v}\left(\vec{a}, \vec{Y}\right) \prod_{\theta=\pm} z_{\rm h}\left(\vec{a}, \vec{Y}, a_t + \theta a_0\right) z_{\rm h}\left(\vec{a}, \vec{Y}, a_1 + \theta a_\infty\right)\right). \tag{C2}$$

Physically, $a$ corresponds to the VEV of the scalar in the vector multiplet, $\epsilon_i$ are two $\Omega$-background parameters regulating the infrared divergence in the localization computation, and $a_0, a_\infty, a_1, a_t$ are related to the masses $m_i$ of the hypermultiplets via

$$m_1 = a_t + a_0, \quad m_2 = a_t - a_0, \quad m_3 = a_1 + a_\infty, \quad m_4 = a_1 - a_\infty. \tag{C3}$$

This function takes the form of a convergent series expansion in $t$,

$$F = \sum_{n \geq 1}^{\infty} c_n(a, a_0, a_t, a_1, a_\infty) t^n, \tag{C4}$$

where the $c_n$ coefficients are rational functions defined via (C2). For example we have

$$c_1(a, a_0, a_t, a_1, a_\infty) = \frac{\left(4a^2 - 4a_0^2 + 4a_t^2 - 1\right)\left(4a^2 + 4a_1^2 - 4a_\infty^2 - 1\right)}{8 - 32a^2}. \tag{C5}$$

The full NS function $F^{\mathrm{NS}}$ includes, on top of the instanton part $F$, the classical and one-loop parts. We have

$$
\begin{aligned}
F^{\mathrm{NS}} =& F - a^2 \log t - \psi^{(-2)}\left(-a - a_0 - a_t + \frac{1}{2}\right) - \psi^{(-2)}\left(a - a_0 - a_t + \frac{1}{2}\right) - \psi^{(-2)}\left(-a + a_0 - a_t + \frac{1}{2}\right) \\
& - \psi^{(-2)}\left(a + a_0 - a_t + \frac{1}{2}\right) - \psi^{(-2)}\left(-a - a_1 - a_\infty + \frac{1}{2}\right) - \psi^{(-2)}\left(a - a_1 - a_\infty + \frac{1}{2}\right) \\
& - \psi^{(-2)}\left(-a - a_1 + a_\infty + \frac{1}{2}\right) - \psi^{(-2)}\left(a - a_1 + a_\infty + \frac{1}{2}\right) \\
& + \psi^{(-2)}(2a + 1) + \psi^{(-2)}(1 - 2a) ,
\end{aligned}
\tag{C6}
$$

where $\psi^{(-2)}(x)$ is the polygamma function of negative order, $\psi^{(-n)}(x) = \frac{1}{(n-2)!}\int_0^z dt\,(z - t)^{n-2}\log\Gamma(t)$ .

## Appendix D: $\mathcal{O}(\mu^2)$ OPE data of double-twist operators

Here we display the results for the OPE data at order $\mu^2$. These expressions are in full agreement with [59] at order $1/\ell^2$, and provide new predictions at higher orders in $1/\ell$. We find

$$
\begin{aligned}
\gamma_{n\ell}^{(2)} =& -\frac{\left((\Delta - 1)\Delta + 6(\Delta - 1)n + 6n^2\right)^2}{8(\ell + 1)^3} - \frac{n(\Delta + n - 2)(\Delta + 2n - 2)^2}{2(\ell + 2)} - \frac{(n + 1)(\Delta + n - 1)(\Delta + 2n)^2}{2\ell} \\
& + \frac{(\Delta - 1)\Delta(8\Delta + 1) + 65n^4 + 130(\Delta - 1)n^3 + (3\Delta(27\Delta - 43) + 133)n^2 + (\Delta - 1)\left(16\Delta^2 + \Delta + 68\right)n}{16(\ell + 1)} \\
& - \frac{(n - 1)n(\Delta + n - 3)(\Delta + n - 2)}{32(\ell + 3)} - \frac{(n + 1)(n + 2)(\Delta + n - 1)(\Delta + n)}{32(\ell - 1)},
\end{aligned}
\tag{D1}
$$

$$
\begin{aligned}
c_{n\ell}^{(2)} =& \frac{1}{8}(\Delta - 2)(9\Delta - 44) - \frac{(2n + 3)(\Delta + n - 1)(\Delta + n)}{32(\ell - 1)} - \frac{3(\Delta + 2n - 1)\left((\Delta - 1)\Delta + 6n^2 + 6(\Delta - 1)n\right)}{4(\ell + 1)^3} \\
& + \frac{(\Delta + 2n - 1)\left(\Delta(16\Delta - 71) + 130n^2 + 130(\Delta - 1)n + 212\right)}{32(\ell + 1)} - \frac{(\Delta + n - 1)(\Delta + 2n)(\Delta + 4n + 2)}{2\ell} \\
& - \frac{(n - 1)n(2\Delta + 2n - 5)}{32(\ell + 3)} - \frac{n(\Delta + 2n - 2)(3\Delta + 4n - 6)}{2(\ell + 2)} + \frac{1}{4}(\psi^{(0)}(n + \ell + 2) - \psi^{(0)}(n + \ell + \Delta)) \\
& \times \left(9\Delta^2 - \frac{89\Delta}{2} + \frac{\left((\Delta - 1)\Delta + 6n^2 + 6(\Delta - 1)n\right)^2}{2(\ell + 1)^3} - \frac{3(\Delta + 2n - 1)\left((\Delta - 1)\Delta + 6n^2 + 6(\Delta - 1)n\right)}{(\ell + 1)^2}\right. \\
& + \frac{(\Delta - 1)(\Delta(4\Delta - 73) + 36) - 65n^4 - 130(\Delta - 1)n^3 + (3(67 - 27\Delta)\Delta - 493)n^2 - (\Delta - 1)(\Delta(16\Delta - 71) + 428)n}{4(\ell + 1)} \\
& + (18\Delta - 89)n + \frac{2n(\Delta + n - 2)(\Delta + 2n - 2)^2}{\ell + 2} + \frac{2(n + 1)(\Delta + n - 1)(\Delta + 2n)^2}{\ell} + \frac{(n - 1)n(\Delta + n - 3)(\Delta + n - 2)}{8(\ell + 3)} \\
& \left. + \frac{(n + 1)(n + 2)(\Delta + n - 1)(\Delta + n)}{8(\ell - 1)} + \left(9\Delta - \frac{71}{2}\right)\ell + 9\right) + \left(\Delta(\Delta + 2) + 6n^2 + 6n(\Delta + \ell) + 3\ell^2 + 3(\Delta + 1)\ell\right)^2 \\
& \times \frac{(\psi^{(0)}(n + \ell + 2) - \psi^{(0)}(n + \ell + \Delta))^2 + \psi^{(1)}(n + \ell + \Delta) - \psi^{(1)}(n + \ell + 2)}{8(\ell + 1)^2} .
\end{aligned}
\tag{D2}
$$

Similar expressions up to order $\mu^5$ can be found in the supplemental files.

## Appendix E: The imaginary part of quasi-normal modes

In this appendix we spell out some details for the computation of (52). The condition for a pole in $G_R(\omega, \ell)$ follows from (26) and reads

$$t^{-2a}e^{\partial_a F}\left(\frac{\Gamma(2a)\Gamma\left(-a-a_t+\frac{1}{2}\right)}{\Gamma(-2a)\Gamma\left(a-a_t+\frac{1}{2}\right)}\right)^2 - \frac{\Gamma\left(a+a_1-a_\infty+\frac{1}{2}\right)\Gamma\left(a+a_1+a_\infty+\frac{1}{2}\right)}{\Gamma\left(-a+a_1-a_\infty+\frac{1}{2}\right)\Gamma\left(-a+a_1+a_\infty+\frac{1}{2}\right)} = 0 \ . \tag{E1}$$

By using the ansatz (51) as well as the dictionary in Table I and the perturbative solution for the real part (43), we obtain

$$\text{Im}\left(\frac{\Gamma\left(a+a_1-a_\infty+\frac{1}{2}\right)\Gamma\left(a+a_1+a_\infty+\frac{1}{2}\right)}{\Gamma\left(-a+a_1-a_\infty+\frac{1}{2}\right)\Gamma\left(-a+a_1+a_\infty+\frac{1}{2}\right)}\right) = \mu^{\ell+1/2}\left(\frac{\Gamma(n+1)\Gamma(n+\Delta-1)}{\Gamma(\ell+n+2)\Gamma(\ell+n+\Delta)}\frac{(-1)^\ell}{3\omega_{n\ell}^{(0)}-2\gamma_{n\ell}^{(1)}}f_{n\ell}^{(1)}+\mathcal{O}(\mu)\right)$$

$$\text{Im}\left(t^{-2a}e^{\partial_a F}\left(\frac{\Gamma(2a)\Gamma\left(-a-a_t+\frac{1}{2}\right)}{\Gamma(-2a)\Gamma\left(a-a_t+\frac{1}{2}\right)}\right)^2\right) = -\mu^{\ell+1/2}\left(\frac{\Gamma\left(\frac{\ell}{2}+1\right)^4}{\Gamma(\ell+1)^2\Gamma(\ell+2)^2}\frac{(-1)^\ell\omega_{n\ell}^{(0)}}{3\omega_{n\ell}^{(0)}-2\gamma_{n\ell}^{(1)}}+\mathcal{O}(\mu)\right) \ , \tag{E2}$$

leading to (52).

## Appendix F: The large $\ell$/large k, fixed $\omega$ limit

Let us consider the limit where $\ell$ is the only large parameter. On the gauge theory side this means that the VEV of the scalar $a$ is much larger than all other parameters. In this limit one can use Zamolodchikov's formula for the Virasoro conformal blocks [110] and the AGT correspondence [8] to show that [103]

$$F = a^2\left(\log\frac{t}{16} + \pi\frac{K(1-t)}{K(t)}\right) + \left(a_1^2 + a_t^2 - \frac{1}{4}\right)\log(1-t) + 2\left(a_0^2 + a_t^2 + a_1^2 + a_\infty^2 - \frac{1}{4}\right)\log\left(\frac{2}{\pi}K(t)\right) + \mathcal{O}(a^{-2}). \tag{F1}$$

Here $K(t)$ is the complete elliptic integral of the first kind. Solving the Matone relation (21) for $a$, we find

$$a = -\frac{(\ell+1)\sqrt{1-2t}K(t)}{\pi} + \mathcal{O}(\ell^{-1}). \tag{F2}$$

Using the asymptotic behavior (F2), we can investigate the behavior of $G_R$ at large $\ell$. We start with the real part of $G_R$, for which the leading behavior comes from the $\sigma = 1$ terms in (26). Expanding at large $a$, we find

$$\text{Re}\, G_R(\omega, \ell) \approx (1+R_+^2)^{\Delta-2}\frac{\Gamma(2-\Delta)}{\Gamma(\Delta-2)}e^{-\partial_{a_1}F}(-a)^{4a_1} \approx \frac{\Gamma(2-\Delta)}{\Gamma(\Delta-2)}\left(\frac{\ell}{2}\right)^{2(\Delta-2)} \tag{F3}$$

Note that this is independent of the temperature.

Now let us turn to the imaginary part. The leading contribution comes from expanding to first order in the $\sigma = -1$ term in both the numerator and denominator of (26). We find

$$\text{Im}\, G_R(\omega, \ell) \approx -\frac{2(1+R_+^2)^{2a_1}e^{\partial_a F-\partial_{a_1}F}t^{-2a}\sin(2\pi a)\sin(2\pi a_1)}{\cos(2\pi(a-a_1))+\cos(2\pi a_\infty)}\frac{\Gamma(2a)^2\Gamma(-2a_1)\Gamma\left(\frac{1}{2}-a+a_1-a_\infty\right)\Gamma\left(\frac{1}{2}-a+a_1+a_\infty\right)}{\Gamma(-2a)^2\Gamma(2a_1)\Gamma\left(\frac{1}{2}+a-a_1-a_\infty\right)\Gamma\left(\frac{1}{2}+a-a_1+a_\infty\right)}$$

$$\times \text{Im}\left(\frac{\Gamma\left(\frac{1}{2}-a-a_t\right)^2}{\Gamma\left(\frac{1}{2}+a-a_t\right)^2}\right)$$

$$\approx -\frac{\Gamma(-2a_1)}{\Gamma(2a_1)}(1+R_+^2)^{2a_1}e^{\partial_a F-\partial_{a_1}F}t^{-2a}2^{8a+1}(-a)^{4a_1}\sin(2\pi a_1)\sinh(2\pi|a_t|), \tag{F4}$$

where in the second equality we took the large $a$ limit. Plugging in the asymptotic behavior (F2) and the dictionary given in Table II gives

$$\text{Im}\, G_R(\omega, \ell) \approx \frac{2\pi\sinh(\pi\omega\sqrt{t(1-2t)})}{\Gamma(\Delta-1)\Gamma(\Delta-2)}\left(\frac{\ell}{2}\right)^{2(\Delta-2)}\exp\left(-2(\ell+1)\sqrt{1-2t}K(1-t)\right). \tag{F5}$$

We see that the imaginary part decays exponentially with spin.

To compute the large $|\mathbf{k}|$ behavior for the black brane, we can take the infinite temperature limit of (F3) and (F5). Using the definition (20) of the brane two-point function, we find

$$G_R^{\text{brane}}(\omega, |\mathbf{k}|) \approx \frac{\Gamma(2-\Delta)}{\Gamma(\Delta-2)}\left(\frac{|\mathbf{k}|}{2}\right)^{2(\Delta-2)} + i\frac{2\pi\sinh\left(\frac{\omega}{2}\right)}{\Gamma(\Delta-1)\Gamma(\Delta-2)}\left(\frac{|\mathbf{k}|}{2}\right)^{2(\Delta-2)}\exp\left(-\sqrt{\frac{\pi}{2}}\frac{|\mathbf{k}|}{\Gamma\left(\frac{3}{4}\right)^2}\right). \qquad \text{(F6)}$$

The rate of exponential decay of the imaginary part matches the result from [29].

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
