# Peer review of "Holographic thermal correlators from supersymmetric instantons"

_SciPost Physics_

## Round 2 · Referee Report · Anonymous (Referee 1) · 2022-11-8

Strengths

1 - The first exact result of a thermal two point function in a four dimensional holographic CFT.

2- Technically sound and extremely interesting.

Weaknesses

1- The organization of the paper could be better, in particular more details in the main text (instead of the long appendices) might be more helpful for a smooth reading.

Report

The paper studies the exact two point function of the thermal holographic CFT in 4d dual to a 5d AdS black hole and to my knowledge this is the first exact result of the same in d>2. The techniques employed to find the exact result is as interesting as the result itself. In particular, identifying the parameters appearing in the wave equation to the masses of the hypermultiplets and solving the nonmalizable and non-normalizable modes using the Nekrasov-Shatashvili free energy turns out to be an extremely efficient way. I definitely recommend publication of this work in SciPost.

Requested changes

However, I would like to mention few small things -

(1) Following (21), I think it will be nice if the authors elaborate a bit on the (non) analyticity of the NS function near half integer values of a and whether it has any bearing on the analytic properties of the retarded thermal two point function. 

(2) Connecting to the section on the heavy-light conformal bootstrap, it will be great if the authors shed some light on the possible usefulness of this formalism in understanding the late time thermalization.

(3) The authors might consider bringing some of the expressions from the appendices to the main text as that might help in a smooth reading of the paper. For instance, I think, some of the expressions given in appendix C would perhaps fit better if incorporated in section 3.

---

## Round 2 · Referee Report · Anonymous (Referee 2) · 2022-11-12

Strengths

1. Very interesting result, a new way to think about the finite-temperature Green's function in a holographic CFT in a regime where it is not controlled by any obvious symmetries

2. Possibly a new way to extract useful data for the bootstrap.

Weaknesses

See below (no real weaknesses, some possible requests for changes).

Report

I found this to be an extremely interesting paper; I (and I think most practitioners) have always thought that there would never be a closed form solution for the wave equation in thermal AdS. In this paper, though the authors have not quite found a completely *closed* form solution, they have found something very close, in that the answer can be written as a compact sum over known functions. Even more interestingly, these known functions are related to results in SUSY gauge theory and can be given a physical meaning. The authors explore some of these physical applications, making interesting connections to previous studies of similar observables. I think particularly interesting are the results in Section 5, where the results are compared to data from the bootstrap. It seems clear that this is only a beginning and there are many other things that could be studied in this formalism. If the correspondence works for higher spins then I am particularly intrigued by the idea of understanding the hydrodynamic regime of thermal correlators from the SUSY gauge theory perspective.

I think this paper is definitely suitable for publication. I have a few relatively minor requests that the authors may want to consider to improve their exposition:

1. The authors discuss the infalling solution to the wave equation, which maps to the retarded propagator. The mathematical significance of this is discussed around (22), but as far as I can tell the physics is not discussed; it would be nice to understand what the infalling/outgoing condition maps to physically in the SUSY gauge theory.

2. The authors do not seem to make a strong statement about why this duality is possible; is it simply a mathematical curiosity (which is my current feeling), or is there some deeper reason as to why these two different systems have the same solution?

3. Finally, the structure of the paper could be made somewhat easier to read, as some rather key points are understood only the in the Appendix -- e.g. of the three regimes discussed in the conclusion, two of them have calculations only in the Appendices. I think transferring some of the results from the Appendix back to the bulk of the paper would make the flow better.

Overall however these are very minor points and I am happy to recommend the paper for publication.

Requested changes

See above.

---

## Editorial Decision

resubmitted